# Cathelicidins Limit Intracellular *Neospora caninum*-Infection in Macrophages

**DOI:** 10.3390/pathogens14070663

**Published:** 2025-07-05

**Authors:** Franco Fiorani, Priyoshi Lahiri, Rodrigo Puentes, Peter John Bradley, Dadin Prando Moore, Eduardo Ruben Cobo

**Affiliations:** 1Faculty of Agricultural Sciences, National University of Mar Del Plata, Balcarce B7620, Argentina; fiorani.franco@inta.gob.ar (F.F.); moore.dadin@inta.gob.ar (D.P.M.); 2Faculty of Veterinary Medicine, University of Calgary, Calgary, AB T2N 4N1, Canada; priyoshi.lahiri@ucalgary.ca (P.L.); rodrigo.puentes@ucalgary.ca (R.P.); 3Department of Microbiology, Immunology and Molecular Genetics, University of California, Los Angeles, CA 90095, USA; pbradley@ucla.edu; 4Institute of Innovation for Agricultural Production and Sustainable Development (IPADS), Balcarce B7620, Argentina

**Keywords:** *Neospora caninum*, cathelicidins, macrophages, host defense peptides

## Abstract

Infections with the protozoan *Neospora caninum* cause abortion in cattle, likely due to the parasite’s replication and excessive inflammation in the placenta. Cathelicidins are host defense peptides known for their antimicrobial and immunomodulatory functions, but their role in *N. caninum* infections remains elusive. Using bone marrow-derived macrophages (BMDMs) isolated from mice expressing (wild-type, *Camp*^+^^/+^) and lacking (*Camp*^−^^/−^) cathelicidins, we investigated the role of endogenous cathelicidin in infections with *N. caninum*. We show that *Camp*^−^^/−^ macrophages primed with lipopolysaccharide (LPS) had an increased number of intracellular *N. caninum* tachyzoites, and these macrophages released higher amounts of IL-1β and lactate dehydrogenase (LDH), a marker of cytotoxicity. These findings indicate that cathelicidins contribute to intracellular *N. caninum* control and inflammation by limiting the activation of the inflammasome, particularly under LPS-induced conditions. This insight reveals the immunomodulatory role of cathelicidins in controlling *N. caninum*-associated pathologies.

## 1. Introduction

Infections with cyst-forming apicomplexan *Neospora caninum* cause abortions in livestock worldwide [1]. This disease, neosporosis, is a leading cause of reproductive losses and abortions in cattle [2]. Although the intrinsic abortigenic mechanisms are still elusive, the pregnancy losses and fetal expulsion are likely driven by a disrupted shift in maternal immune cells from a tolerant state to a proinflammatory phenotype [3,4]. The inflammatory response elicited by macrophages could be essential in promoting the inflammatory stage during abortions. The NLR family pyrin domain containing 3 (NLRP3) inflammasome pathway is activated in macrophages in response to related protozoa, *Toxoplasma gondii*, *Plasmodium*, and *Leishmania* [5,6,7]. Moreover, *N. caninum* triggers NLRP3 inflammasome activation in peritoneal macrophages, bone marrow-derived macrophages (BMDMs), and bovine macrophages, leading to caspase-1 activation and secretion of IL-1β and IL-18 [8,9,10,11]. Pyroptosis has also been promoted in macrophages exposed to *N. caninum*, as a lytic and proinflammatory form of programmed cell death [12]. While macrophage responses can be beneficial in limiting the replication of intracellular pathogens, this excessive inflammatory and pyroptotic activity would contribute to placental and fetal damage and abortion [13,14,15,16,17].

Cathelicidins are cationic antimicrobial and immunomodulatory peptides [18,19,20] encoded by the *Camp* gene [21] and produced by multiple species, including mice, chickens, horses, pigs, and cattle [22]. Cathelicidins have been shown to regulate inflammation and tissue damage during infections [23,24,25]. The sole human cathelicidin, LL-37, encoded by the *CAMP* gene, inhibits caspase-1 activation and suppresses inflammasome activation and pyroptosis in macrophages [26]. Moreover, stimulating human THP-1 macrophages with synthetic murine cathelicidin (cathelicidin-related antimicrobial peptide; CRAMP) reduces the load of intracellular *N. caninum* tachyzoites [27] and *Mycobacterium paratuberculosis* [28]. However, the role of endogenous cathelicidin in macrophage defense against intracellular pathogens, like *N. caninum*, remains elusive. In this study, we analyze the *N. caninum* infection in BMDMs isolated from mice deficient in cathelicidins (*Camp*^−^^/−^) to demonstrate that endogenous cathelicidin limits a deleterious inflammatory and inflammasome response in macrophages.

## 2. Materials and Methods

### 2.1. Experimental Design and Animal Models

This study investigated the role of cathelicidins in the innate immune response to *N. caninum* infection using an in vitro macrophage model. BMDMs were isolated from male 8-week-old wild-type *Camp*^+/+^ and cathelicidin-null *Camp*^−^^/−^ C57BL/6 mice (B6.129X1-Camp^tm1Rlg/J^; The Jackson Laboratory, Bar Harbor, ME, USA) [21]. Mice were housed in a pathogen-free environment provided by the University of Calgary, and humanely euthanized to collect bone marrow. The present study was conducted following regulations specified by the Canadian Guidelines for Animal Welfare (CGAW) and was approved by the University of Calgary Health Sciences Animal Care Committee (AC24-0045).

*Camp*^+^^/+^ and *Camp*^−^^/−^ BMDMs were seeded separately on 24-well plates and were either left untreated (RPMI control) (*n*: 4) or infected with *N. caninum* tachyzoites (*n*: 4). Previous studies have demonstrated the essential role of lipopolysaccharide (LPS) pretreatment in *N. caninum*-induced IL-1β release in BMDM supernatants [9]. Therefore, a group stimulated with LPS (*n:* 4) or LPS-primed and *N. caninum*-infected BMMDs (*n*: 4) were added in this study. All treatments were applied to both *Camp*^+^^/+^ and *Camp*^−^^/−^ BMDMs and analyzed at 12 and 24 h post-infection (hpi). It was a 2 × 4 factorial design (two genotypes: *Camp*^+^^/+^ and *Camp*^−^^/−^; and had four treatment conditions: Control (Roswell Park Memorial Institute 1640 medium (RPMI)), LPS only, *N. caninum* only, and LPS + *N. caninum*, resulting in eight experimental groups. The experimental design is shown (Figure 1).

To assess the impact of cathelicidin deficiency on infection and inflammation, BMDM supernatants were removed and immediately stored at −20 °C for the measurement of IL-1β secretion and lactate dehydrogenase (LDH) as indicators of inflammasome activation and cell membrane integrity, respectively. Intracellular tachyzoite load was quantified, and confocal microscopy was used to visualize parasite–host cell interactions. Those assays were conducted in technical duplicates and in a blinded manner to reduce observer bias.

### 2.2. Cell Isolation and Culture Protocols

BMDMs were isolated from femurs and tibias aseptically removed from *Camp*^+/+^ and *Camp*^−^^/−^ mice following a previously published protocol [29]. Upon isolation, cells were counted and seeded separately on 24-well plates at 2.3 × 10^5^ cells/well and inspected daily by light microscopy. Cells were cultures for 6 days for macrophage lineage differentiation in RPMI supplemented with 10% fetal bovine serum (FBS), 2 mM L-glutamine, 50 μM 2-mercaptoethanol, 10 mM HEPES buffer (pH 7.4), 1% penicillin (100 U mL^−1^)/streptomycin (100 μg mL^−1^), and 10% conditioned media from L929 cells as a source of macrophage colony-stimulating factor. The medium was changed to complete medium (RPMI) plus 1% FBS for experiments. BMDMs were pre-treated with LPS (100 ng/mL; 3 h) [9], washed twice with PBS, and infected with *N. caninum* at multiplicities of infection (MOI) 3:1 (parasite–cell) for 12 and 24 hpi. BMDMs treated with medium alone were negative controls. An additional group of BMDMs was infected with *N. caninum* (MOI 3:1) without prior LPS treatment.

### 2.3. Parasite Preparation and Infection Procedures

*N. caninum* tachyzoites (Liverpool strain, Nc Liv) constitutively expressing green fluorescent protein (GFP) were utilized in this study [30]. Parasites were maintained by serial passages in human foreskin fibroblasts (HFFs) in Dulbecco’s Modified Eagle Medium (DMEM) supplemented with 10% FBS, 100 U/mL penicillin, and 100 µg/mL streptomycin and incubated at 37 °C in a humidified atmosphere containing 5% CO_2_. For parasite purification, infected monolayers were washed with cold phosphate-buffered saline (PBS), detached by trypsinization, and subjected to mechanical disruption using repeated passages through a 27-G needle. The suspension was passed through a 5.0 μm pore-size filter (Millipore Sigma, Darmstadt, Germany) to remove host cell debris. Purified tachyzoites were washed twice with RPMI-1640, counted using a hemocytometer, and subsequently used to infect BMDMs.

### 2.4. Quantification of Intracellular Parasite Load

To quantify intracellular *N. caninum* tachyzoites in *Camp*^+^^/+^ and *Camp*^−^^/−^ BMDMs at 12 and 24 hpi, infected cell monolayers were gently washed with PBS and scraped using a sterile plastic cell scraper. The resulting suspension was subjected to mechanical disruption by repeated passage through a 27-G needle to release intracellular parasites. To eliminate host cell debris, the suspension was passed through a 5.0-μm pore-size filter (Millipore Sigma, Darmstadt, Germany). Tachyzoites were then counted using trypan blue exclusion in a Neubauer hemocytometer. Briefly, 10 μL of the suspensions were loaded onto a cleaned hemocytometer. Tachyzoites were counted under a light microscope at 40 × magnification in five large squares, applying standard inclusion rules. The mean number of tachyzoites per square was calculated, and parasite concentration was estimated using the following formula: average count per square × 10^4^ × dilution factor. Results were expressed as the number of viable tachyzoites/mL.

### 2.5. Confocal Microscopy

BMDMs were plated on coverslips in 6-well plates, pre-treated with LPS (100 ng/mL; 3 h), and stimulated with GFP Nc Liv tachyzoites (MOI  =  3) for 12 and 24 h. Cells were rinsed with PBS and fixed in formaldehyde solution 3.7% (*v*/*v*), rinsed in cold PBS, blocked with PBS and 1% (*w*/*v*) bovine serum albumin (20 min, RT), and rinsed with PBS. Cells were incubated with Alexa Fluor^®^ 568 phalloidin (A12380, Thermo Fisher Scientific, Waltham, MA, USA) (20 min, RT), rinsed in cold PBS, and blotted (30 min, RT) with nuclei counterstained with 4′,6-diamidino-2-phenylindole (DAPI) (Thermo Fisher). Sections were rinsed with cold PBS and mounted with ProLong^®^ Gold reagent (Thermo Fisher). Slides were examined using a FluoView FV1000 confocal immunofluorescence microscope (Olympus, Tokyo, Japón).

### 2.6. Cytokine and Lactate Dehydrogenase Determinations

BMDM culture supernatants were collected at 12 and 24 hpi to analyze cytokine secretion and cell death via lactate dehydrogenase (LDH) release (Roche Diagnostics) (Figure 1). Lactate dehydrogenase release was expressed as a percentage, calculated using the following formula: (LDH from infected samples–LDH from uninfected controls)/(LDH from total cell lysis–LDH from uninfected controls) × 100. The remaining supernatants were aliquoted and stored at −20 °C until determination of secreted IL-1β by murine-specific enzyme-linked immunosorbent assay (ELISA) (eBioscience^TM^, Thermo Fisher Scientific, Waltham, MA, USA). 

### 2.7. Statistical Analysis

Data were analyzed using Prism 5.0 software (GraphPad Software, Inc., Boston, MA, USA) and presented as mean ± standard deviation (SD). The D’Agostino–Pearson test was used to assess the normality of the data. Differences between groups were determined using analysis of variance (ANOVA) and Tukey’s test for data with residuals of normal distribution and homoscedastic variances, as observed in Bartlett’s test. A *p*-value <  0.05 was considered statistically significant.

## 3. Results

### 3.1. Cathelicidin Deficiency Modulates Intracellular N. caninum Burden in Macrophages

To assess the impact of cathelicidin on *N. caninum* infection, BMDMs from *Camp*^+/+^ and *Camp*^−/−^ mice were challenged with *N. caninum*. *Camp*^−/−^ macrophages pre-treated with LPS showed a higher number of intracellular tachyzoites than the *Camp*^+/+^ counterparts at 12 hpi (*p* < 0.05) (Figure 2). *N. caninum* load in *Camp*^+/+^ macrophages exposed to LPS was elevated compared to *Camp*^+/+^ macrophages infected by *N. caninum* without previous LPS (*p* < 0.01) (Figure 2). *Camp*^+/+^ and *Camp*^−/−^ macrophages without LPS stimulation performed similarly (*p* > 0.05) (Figure 2). A similar response was observed at a late stage, 24 hpi, with the highest parasite burden in infected *Camp*^−/−^ macrophages pre-treated with LPS (*p* > 0.05) (Figure 2). Macrophages without LPS also exhibited higher parasite loads than the *Camp*^+/+^ counterparts (*p* < 0.0001) (Figure 2). Using confocal microscopy, we confirmed increased intracellular parasite burden in *Camp*^−/−^ macrophages. A higher intracellular GFP-positive *N. caninum* tachyzoite burden was visualized in LPS-primed *Camp*^−/−^ macrophages than in the *Camp*^+/+^ counterparts (Figure 3).

### 3.2. Cathelicidin Deficiency Exacerbates the Secretion of IL-1β and Cell Damage in Response to N. Caninum

Under basal conditions and in the presence of *N. caninum*, secreted IL-1β levels remained comparably low in *Camp*^+/+^ and *Camp*^−/−^ macrophages (*p* > 0.05) (Figure 4). In contrast, in LPS-primed macrophages, released IL-1β production was higher in *Camp*^−/−^ macrophages than the *Camp*^+/+^ counterparts at 12 h (*p* < 0.0001) (Figure 4). Secreted IL-1β response in LPS-primed *Camp*^−/−^ macrophages was higher that the *Camp*^+/+^ counterparts at 24 hpi with *N. caninum* (Figure 4). Levels of IL-1β dropped between 12 and 24 hpi in *Camp*^+/+^ macrophages, but remained high in *Camp*^−/−^ macrophages (*p* < 0.0001).

To evaluate the cell membrane integrity and cytotoxicity of these macrophages exposed to *N. caninum*, the lactate dehydrogenase (LDH) released by *N. caninum*-infected BMDMs was quantified. In unstimulated conditions, regardless of infection with *N. caninum*, LDH levels remained comparatively low in *Camp*^+/+^ and *Camp*^−/−^ macrophages (*p* > 0.05), indicating minimal baseline cytotoxicity (Figure 4). LPS-primed *Camp*^−/−^ macrophages secreted higher amounts of LDH than the *Camp*^+/+^ counterparts in sham conditions (*p* < 0.0001) and after being challenged by *N. caninum* (*p* < 0.05) (Figure 4).

## 4. Discussion

This study determined a critical role for endogenous cathelicidins in the innate immune response to *N. caninum* infection. Using primary BMDMs, which are physiologically relevant as they mimic real-life immune responses, we show that cathelicidins limit the intracellular parasite burden and secretion of IL-1β, while preventing exaggerated cytotoxicity in macrophages. The modulatory effects of cathelicidins were particularly evident under LPS-induced inflammatory conditions, indicating that these peptides may help limit parasite invasion or regulate inflammation during active infection processes.

Our observations suggest that the endogenous synthesis of cathelicidin encompasses not only antimicrobial activities but also regulates the inflammasome. The inflammasome is a complex machinery that produces active caspase-1, the enzyme that cleaves pro-IL-1β into active IL-1β and triggers an inflammatory form of programmed cell death: pyroptosis [30]. We demonstrated that the absence of cathelicidin in LPS-primed macrophages resulted in elevated levels of IL-1β in response to *N. caninum*. Furthermore, the *Camp*^+/+^ macrophages reduced IL-1β synthesis by 24 hpi, while *Camp*^−/−^ macrophages maintained elevated cytokine production. Additionally, there was increased LDH release in *Camp*^−/−^ macrophages, indicating cell membrane damage linked to pyroptosis. Consistently, human cathelicidin (LL-37) decreased IL-1β levels and enhanced survival in murine bacterial sepsis models [31]. The role of cathelicidin in regulating the inflammasome and limiting inflammasome-driven cytotoxicity is crucial, as excessive inflammasome activation is associated with pathological and chronic inflammation. Our observed higher intracellular *N. caninum* tachyzoite burden in *Camp*^−/−^ macrophages also indicates that cathelicidin is essential to limit protozoan infection. Cathelicidins have been shown to impair the intracellular survival of protozoal pathogens, including *Leishmania* and *N. caninum* [6,27,32]. Endogenous cathelicidin limited cutaneous infections with *Leishmania* [33], while equine cathelicidin (eCath1) inhibited in vitro the survival of a bloodstream form of trypomastigotes from related protozoa (*Trypanosoma brucei brucei*, *Trypanosoma evansi*, and *Trypanosoma equiperdum*) through structural alterations to the parasite’s membrane [34]. Thus, cathelicidins may act as negative regulators of sustained inflammation, promoting its timely resolution. Additionally, their antiprotozoal effects may involve immunomodulatory and microbicidal roles in the host. Regarding neosporosis and reproductive losses, placental cells express a variety of antimicrobial peptides, including cathelicidin [35]. The source of cathelicidin expression originates from trophoblasts and infiltrating leukocytes [35], such as neutrophils, which store larger quantities of this peptide in secondary granules [36,37,38]. Thus, the influx of leukocytes during placentitis and the release of endogenous cathelicidin could limit the damage to the placenta and fetus, as well as the *N. caninum* infection. Conversely, the loss of this peptide would contribute to uncontrolled maternal inflammation, characterized by increased production of interferon-γ (IFN-γ) and heightened cytotoxic activity of CD4^+^ T cells [39,40,41] and, eventually, abortions in *N. caninum*-infected cattle [42]. While the small sample size limited statistical power to detect more significant biological differences, this in vitro model provides mechanistic insight into the macrophage response in neosporosis. Still, studies should focus on the complexity of the placenta and the maternal–fetal interface to further understand the role of cathelicidins in systemic and reproductive pathogenesis during *N. caninum* infection. In summary, cathelicidins are essential regulators of the macrophage response to *N. caninum*, limiting parasite burden and inflammasome-mediated inflammation. These innate effectors, such as cathelicidin, may maintain immune balance in the placenta during neosporosis, controlling *N. caninum*-induced immunopathology and abortions.

## Figures and Tables

**Figure 1 pathogens-14-00663-f001:**
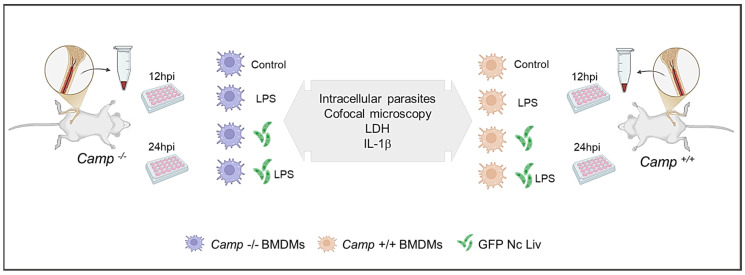
Schematic representation of the experimental design. Bone marrow-derived macrophages (BMDMs) from *Camp*^+/+^ (wild-type) and *Camp*^−/−^ (cathelicidin-deficient) mice were cultured and assigned to four experimental conditions: untreated (Control), lipopolysaccharide (LPS) stimu-lation, green fluorescent protein (GFP)-*N. caninum* Liverpool (Nc Liv) tachyzoites infection (Nc), and LPS pre-treatment followed by GFP *N. caninum* infection (Nc). Supernatants and cell lysates were collected at 12 and 24 h post-infection (hpi) for lactate dehydrogenase (LDH) cytotoxicity assay, IL-1β ELISA, and quantification of intracellular tachyzoites. Each group included four biological replicates. Confocal microscopy was performed to visualize the parasites.

**Figure 2 pathogens-14-00663-f002:**
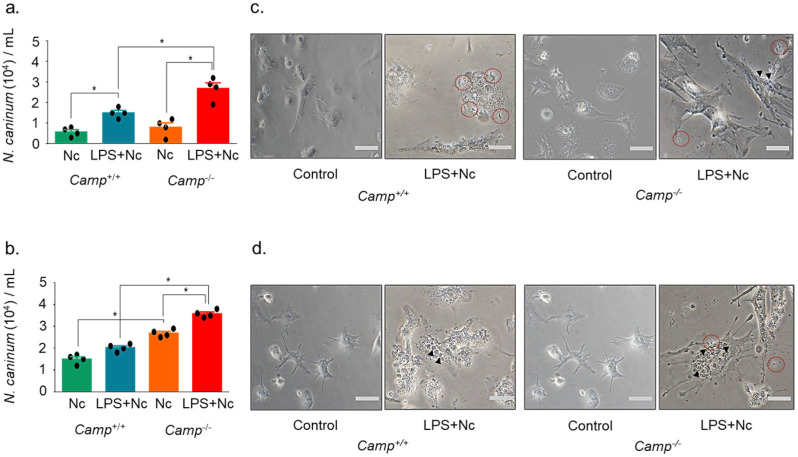
*N. caninum* tachyzoite burden in *Camp*^+/+^ and *Camp*^−^^/−^ bone marrow-derived macrophages. Bone marrow-derived macrophages (BMDMs) from *Camp*^+/+^ and *Camp*^−/−^ (cathelicidin-deficient) mice were left untreated (Control) or subjected to lipopolysaccharide (LPS) stimulation, *N. caninum* infection (Nc), or LPS pre-treatment followed by infection (LPS + Nc). Intracellular tachyzoites were quantified at 12 (**a**) and 24 (**b**) hours post-infection (hpi). Representative microscopy images of BMDMs at 12 hpi (**c**) and 24 hpi (**d**) illustrate the intracellular parasite burden. Intracellular tachyzoites are indicated by solid triangles, while dashed circles mark extracellular parasites. Data represent the mean ± SD of four independent replicates per condition. * *p* < 0.05 was considered statistically significant. Scale bars = 50 μm.

**Figure 3 pathogens-14-00663-f003:**
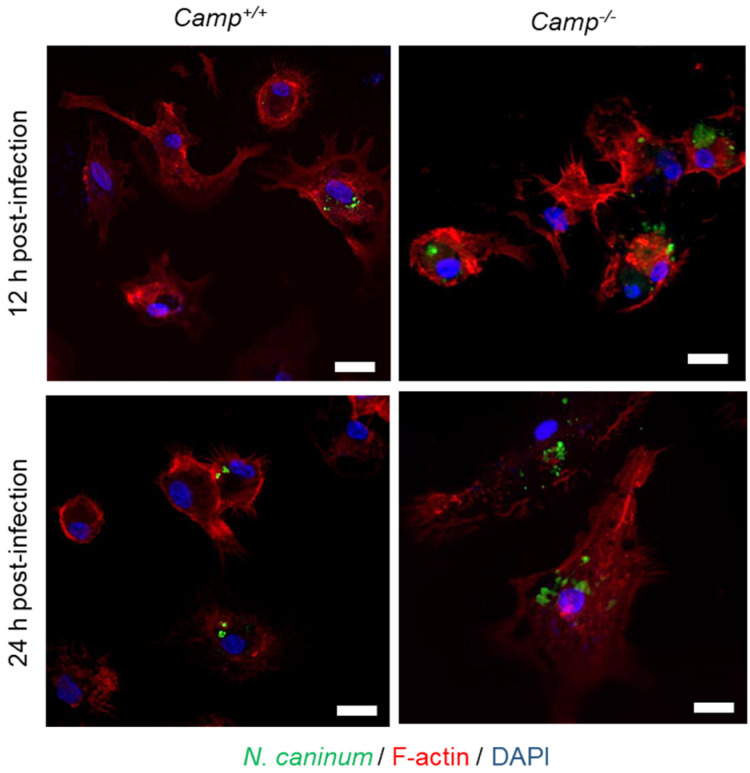
Visualization of intracellular *N. caninum* tachyzoites in *Camp*^+^^/+^ and *Camp*^−^^/−^ bone marrow-derived macrophages. Confocal microscopy images of bone marrow-derived macrophages (BMDMs) from *Camp*^+/+^ and *Camp*^−/−^ (cathelicidin-deficient) mice pre-treated with lipopolysaccharides (LPS) and subsequently infected with GFP-expressing *Neospora caninum* tachyzoites (LPS + Nc). Cells were fixed and stained at 12 and 24 h post-infection (hpi). F-actin filaments were visualized using Alexa Fluor^®^ 568-conjugated phalloidin (red), nuclei were counterstained with 4′,6-diamidino-2-phenylindole (DAPI, blue), and tachyzoites were detected via green fluorescent protein (GFP, green). Scale bar = 50 μm. Representative images illustrate parasite localization within the host cell cytoplasm and associated cytoskeletal structures.

**Figure 4 pathogens-14-00663-f004:**
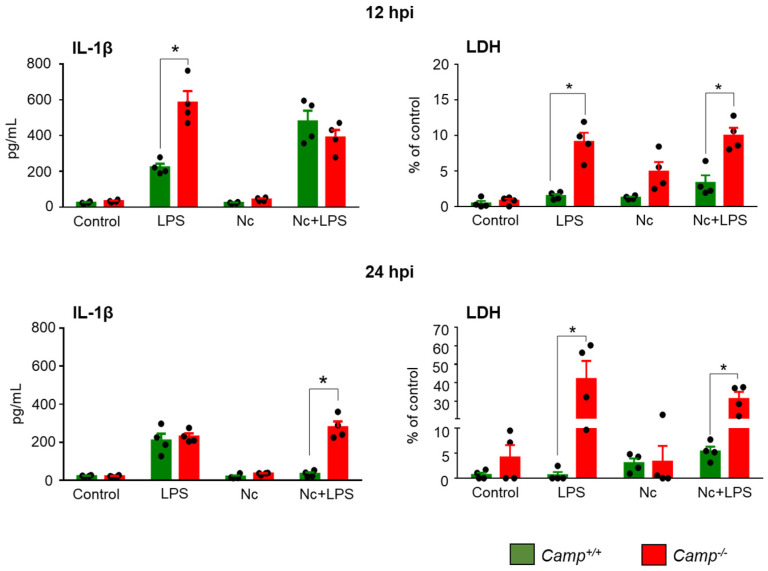
IL-1β secretion and lactate dehydrogenase release in *Camp*^+/+^ and *Camp*^−/−^ bone marrow-derived macrophages exposed to *N. caninum*. Bone marrow-derived macrophages (BMDMs) from *Camp*^+/+^ and *Camp*^−/−^ (cathelicidin-deficient) mice were exposed to one of the following conditions: medium alone (Control), lipopolysaccharide (LPS), *N. caninum* (Nc), or LPS followed by *N. caninum* infection (Nc + LPS). Supernatants were collected at 12 and 24 h post-infection (hpi) to quantify IL-1β and lactate dehydrogenase (LDH) levels using enzyme-linked immunosorbent assay (ELISA). Each experimental condition included four biological replicates (*n* = 4). Data are presented as mean ± standard deviation (SD). A *p*-value < 0.05 was considered statistically significant and is indicated by an asterisk (*).

## Data Availability

The original contributions presented in the study are included in the article; further inquiries can be directed to the corresponding author.

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
