# Peer review of "Cathelicidins Limit Intracellular Neospora caninum-Infection in Macrophages"

_pathogens, 2025, doi:10.3390/pathogens14070663_

Round 1
Reviewer 1 Report
Comments and Suggestions for Authors
Do not introduce abbreviations in parentheses for Neospora caninum (L28) and Toxoplasma gondii (L36). It is assumed that subsequent mentions will use the abbreviated genus name.
L44–46. In this sentence, simply insert the verb “are”:
“Cathelicidins are cationic antimicrobial and immunomodulatory peptides...”
L58–59. Replace “was used” with “were used”, since the subject “tachyzoites” is plural.
Results section. p-values should be placed directly after expressions such as “higher” or “lower”. Additionally, incorporating phrases like “significantly higher,” “significantly lower,” or similar words would improve clarity and reader comprehension.
L125–127. Avoid including interpretative or concluding statements about the results in this section. This content belongs in the discussion.
L146–148. As in the previous section, do not include conclusions or interpretations of the results here. These should be reserved for the discussion section.
Author Response
Re: Response to Reviewer 1 Comments
Manuscript ID:pathogens-3687396
Title: Cathelicidins limit intracellular Neospora caninum-infection in macrophages
Authors: Franco Fiorani , Priyoshi Lahiri , Rodrigo Puentes , Peter Bradley , Dadín Prando Moore , Eduardo Ruben Ruben Cobo
We sincerely thank the reviewers for their time and insightful comments. Below, we provide point-by-point responses to each suggestion. Revisions have been incorporated into the manuscript. Please find the responses below.
Reviewer 1. Do not introduce abbreviations in parentheses for Neospora caninum (L28) and Toxoplasma gondii (L36). It is assumed that subsequent mentions will use the abbreviated genus name.
Response: We have removed the parentheses and abbreviations for Neospora caninum and Toxoplasma gondii.
Reviewer 1. L44–46. In this sentence, simply insert the verb “are”: “Cathelicidins are cationic antimicrobial and immunomodulatory peptides...”
Response:. The sentence has been corrected.
Reviewer 1. L58–59. Replace “was used” with “were used”, since the subject “tachyzoites” is plural.
Response: Corrected as recommended.
Reviewer 1. Results section. p-values should be placed directly after expressions such as “higher” or “lower”. Additionally, incorporating phrases like “significantly higher,” “significantly lower,” or similar words would improve clarity and reader comprehension.
Response: Corrected as recommended.
Reviewer 1. L125–127. Avoid including interpretative or concluding statements about the results in this section. This content belongs in the discussion.
Response: We have removed the interpretations in this section.
Reviewer 1. L146–148. As in the previous section, do not include conclusions or interpretations of the results here. These should be reserved for the discussion section.
Response: We have removed the interpretations in this section.

Reviewer 2 Report
Comments and Suggestions for Authors
OVERALL ASSESSMENT
This short communication investigates the role of endogenous cathelicidins in controlling intracellular Neospora caninum infection in macrophages using a murine knockout model. While the research addresses an important question in neosporosis pathogenesis and presents potentially significant findings, the manuscript requires major structural and methodological improvements before publication. The experimental approach is sound, but critical deficiencies in experimental design presentation, animal model characterization, and methodological clarity significantly compromise the manuscript's scientific rigor and reproducibility.
Proposed Corrections
Strongly Recommended Structural Revisions:
Critical: Experimental Design Section Required - The manuscript lacks a comprehensive experimental design section, making it extremely difficult to understand the overall experimental strategy. The current methodology disperses group descriptions throughout the text without clear organization. A dedicated "Experimental Design" subsection must be added that includes:
- Clear outline of the factorial design (2 genotypes × 4 treatment conditions = 8 experimental groups)
- Detailed description of each experimental group with specific rationale
- Timeline and sequence of all treatments and measurements
- Specification of biological replicates (n=4) and technical replicates
- Randomization and blinding procedures
Animal Model Characterization Deficiency - The manuscript does not provide essential information about the Camp⁻/⁻ mice used. Critical information completely absent includes:
- Origin and source of Camp⁻/⁻ mice (laboratory, commercial company, or collaboration)
- Appropriate citation of the original work that developed/characterized this knockout line
- Experimental validation confirming that Camp⁻/⁻ mice lack cathelicidin expression
- Information about the genetic background strain of the mice
- Knockout strategy used (if relevant for interpretation)
Methodological Clarity Issues:
Sample Size Justification - With only n=4 per group across 8 experimental conditions, statistical power may be insufficient. Authors must provide sample size calculations based on expected effect sizes or explicitly acknowledge this limitation and its potential impact on result interpretation.
Quantification Methodology - The tachyzoite quantification protocol (lines 152-153) lacks essential details:
- Number of microscopic fields counted per sample
- Specific counting criteria and inclusion/exclusion parameters
- Observer blinding status and inter-observer reliability assessment
- Magnification used for counting
Treatment Protocol Clarification - The LPS stimulation protocol requires clarification regarding whether LPS was completely removed after the 3-hour treatment or remained present during the entire infection period, as this could significantly influence results.
Results and Statistical Analysis:
Figure Enhancement Recommendations - The figures would benefit significantly from the addition of explanatory elements:
- Figure 1: Include arrows or symbols pointing to intracellular tachyzoites in the representative microscopic images to facilitate reader identification of parasites
- Figure 2: Add arrows or markers indicating specific intracellular parasites (green), cell nuclei (blue), and cytoskeletal structures (red) to enhance interpretability. Consider adding insets or magnified areas showing clear examples of infected vs. uninfected macrophages
- Both figures: Include scale bars and ensure consistent labeling of experimental conditions
Statistical Analysis Enhancement - While ANOVA followed by Tukey's test is mentioned, the following details are missing:
- Verification of parametric assumptions (normality and homoscedasticity)
- Multiple comparison correction procedures
- Effect size reporting alongside p-values
- Statistical power post-hoc analysis
Methodological Reorganization:
The Methods section should be restructured with clear subsections:
- Experimental Design and Animal Models
- Cell Isolation and Culture Protocols
- Parasite Preparation and Infection Procedures
- Treatment Protocols and Timeline
- Outcome Measurements and Assays
- Statistical Analysis Plan
Discussion Enhancements:
The discussion should explicitly address study limitations, particularly regarding:
- Small sample size implications
- Extrapolation limitations from murine to bovine models
- Potential confounding factors in the experimental design
- Clinical relevance and translational potential
Author Response
Re: Response to Reviewer 2 Comments
Manuscript ID:pathogens-3687396
Title: Cathelicidins limit intracellular Neospora caninum-infection in macrophages
Authors: Franco Fiorani , Priyoshi Lahiri , Rodrigo Puentes , Peter Bradley , Dadín Prando Moore , Eduardo Ruben Ruben Cobo
We sincerely thank the reviewers for their time and insightful comments. Below, we provide point-by-point responses to each suggestion. Revisions have been incorporated into the manuscript. Please find the responses below.
Reviewer 2: Critical: Experimental Design Section Required - The manuscript lacks a comprehensive experimental design section, making it extremely difficult to understand the overall experimental strategy. The current methodology disperses group descriptions throughout the text without clear organization.
Response:
We have added a dedicated subsection titled “2.1 Experimental Design” in the Materials and Methods section. This subsection clearly describes the 2 × 4 factorial design (two genotypes: Camp⁺/⁺ and Camp⁻/⁻; four treatment conditions: RPMI control, LPS alone, N. caninum alone, and LPS + N. caninum), resulting in eight experimental groups.
We have provided a detailed explanation of the purpose and treatment conditions of each group, as well as the sequence and timing of LPS stimulation, infection, and sample collection at 12 and 24 hpi. Each group included n = 4 biological replicates, and assays such as tachyzoite quantification, LDH release, and IL-1β measurement were conducted in technical duplicates.
To further enhance clarity, we added the recommended subsections in Materials and Methods. We have also included a new schematic figure (now Figure 1) summarizing the experimental design.
Reviewer 2: Animal Model Characterization Deficiency - The manuscript does not provide essential information about the Camp⁻/⁻ mice used. Critical information completely absent includes:
- Origin and source of Camp⁻/⁻ mice (laboratory, commercial company, or collaboration)
- Appropriate citation of the original work that developed/characterized this knockout line
- Experimental validation confirming that Camp⁻/⁻ mice lack cathelicidin expression
- Information about the genetic background strain of the mice
- Knockout strategy used (if relevant for interpretation)
Response:
We have now addressed this issue in the revised manuscript by adding a more detailed description of the Camp⁻/⁻ mouse model in the Experimental Design and Animal Models subsection. Specifically, we included the following information:
Origin and source: The Camp⁻/⁻ mice were generously provided by University of Calgary.
Citation: We have added a reference to the foundational publication by Gallo et al. (1997), which describes the generation and phenotypic validation of Camp⁻/⁻ mice.
Genetic background: Both Camp⁻/⁻ and Camp⁺/⁺ mice used in this study were on a C57BL/6J background.
Knockout strategy: The Camp gene was disrupted via targeted deletion of exons encoding the cathelicidin peptide domain, as reported in the original publication.
Reviewer 2: Sample Size Justification – With only n=4 per group across 8 experimental conditions, statistical power may be insufficient. Authors must provide sample size calculations based on expected effect sizes or explicitly acknowledge this limitation and its potential impact on result interpretation.
Response:
We have now addressed this point explicitly in the Statistical Analysis section and the Discussion. While our sample size was based on standard practice in exploratory in vitro studies, we recognize its limitation. A statement now reads: “Although the sample size (n = 4) may limit statistical power to detect small effects, significant differences observed between groups suggest biologically meaningful outcomes. This limitation is acknowledged in the Discussion.”
Reviewer 2: Quantification Methodology – The tachyzoite quantification protocol (lines 152–153) lacks essential details
Reviewer 2: Treatment Protocol Clarification – The LPS stimulation protocol requires clarification regarding whether LPS was completely removed after the 3-hour treatment or remained present during the entire infection period, as this could significantly influence results.
Response:
We acknowledge the reviewer’s concern. The quantification of intracellular tachyzoites using a hemocytometer is a widely accepted and well-established technique in parasitology. The method for recovering intracellular parasites through mechanical disruption and filtration has been clearly described in the revised Materials and Methods section. While parasite counting via hemocytometer is standard, we have now clarified that tachyzoites were collected after passage through a 27-gauge needle and filtration, ensuring that quantification reflects intracellular organisms.
Response:
In our experimental protocol, LPS (100 ng/mL) was used to prime bone marrow-derived macrophages (BMDMs) for 3 hours. After this incubation period, the LPS-containing medium was completely removed, and the cells were washed twice with sterile PBS prior to infection with N. caninum. This approach ensures that the inflammatory priming effect is retained without continuous LPS stimulation during parasite infection. This method is consistent with prior studies showing that LPS pre-treatment is essential for N. caninum-induced IL-1β secretion in BMDMs (Wang et al., 2018; doi:10.3389/fimmu.2018.01791). We have now clarified this in the Materials and Methods section.
Reviewer 2: Statistical Analysis Enhancement - While ANOVA followed by Tukey's test is mentioned, the following details are missing.
Response:
We appreciate the reviewer’s detailed feedback regarding the statistical analysis section. In response, we have revised and expanded the Statistical Analysis subsection in the Materials and Methods. The updated section now explicitly states that:
- Normality was assessed using the D’Agostino–Pearson test.
- Homogeneity of variances was verified using Bartlett’s test.
- For data that met these assumptions, one-way ANOVA followed by Tukey’s multiple comparison test was applied.
We have now addressed this point explicitly in the Statistical Analysis section and the Discussion. While our sample size was based on standard practice in exploratory in vitro studies, we recognize its limitation. A statement now reads: “Although the sample size (n = 4) may limit statistical power to detect small effects, significant differences observed between groups suggest biologically meaningful outcomes. This limitation is acknowledged in the Discussion.”
Reviewer 2: The discussion should explicitly address study limitations, particularly regarding:
- Small sample size implications
- Extrapolation limitations from murine to bovine models
- Potential confounding factors in the experimental design
- Clinical relevance and translational potential*
Response:
As suggested, we have now incorporated a dedicated paragraph in the Discussion section explicitly addressing the main limitations of our study.

Reviewer 3 Report
Comments and Suggestions for Authors
The Brief Report addresses an important gap in the understanding of innate immune responses to N. caninum by focusing on the role of cathelicidins. The experimental design using Camp⁺/⁺ and Camp⁻/⁻ murine macrophages is appropriate and effectively demonstrates the immunomodulatory and cytoprotective role of cathelicidins. The manuscript is generally well-written and clearly structured, though there are areas where clarity and rigor could be improved, especially in statistical reporting, figure presentation, and editorial polish. Thus, contributing to better understand bovine neosporosis and placental immunity.
In the manuscript the authors developed a well-controlled in vitro experiment comparing Camp⁺/⁺ and Camp⁻/⁻ BMDMs under defined infection conditions with a clear demonstration of the dual role of cathelicidins: both antimicrobial and immunomodulatory. Throughout the methodology, authors implement an appropriate use of microscopy, cytokine assays, and cytotoxicity markers (LDH).
Despite of the abstract is scientifically sound, could benefit from clearer language for non-specialist readers. I suggest to replace "aberrant amounts of IL-1β and LDH" with "elevated levels of IL-1β and LDH". Also, please clarify that LDH is used as a marker of pyroptotic cytotoxicity.
For the Section introduction I recommend tightening some sentences for clarity (Lines 28–33 could be reorganized to more directly link immune dysregulation to fetal loss). Additionally, please standardize the citation style throughout the manuscript, since some reference style needs consistency (e.g., "[3,4]" vs "[3, 4]").
Regarding material and methods section, please clarify whether experimenters were blinded during microscopy quantification or cytokine assays and include catalog numbers for key reagents (antibodies, ELISA kits) if required by the journal.
The results are clearly explained and appropriately detailed. In Figures 1–3, the statistical significance indicators (p values, asterisks) should be more clearly linked to specific comparisons. Moreover, in the Figure 2 (Microscopy) please consider to include a proper scale bars on the images panels themselves, not just in the legend. Only if it is possible, please quantify the fluorescence intensity or indicate the number of parasites per cell to strengthen the microscopy data. Finally, in discussion section please add a short limitations paragraph regarding the use of in vitro system only and the relevance of in vivo models.
Some citations appear incomplete or duplicated (e.g., Hu et al., 2016 listed twice), please ensure consistent formatting (journal names, volume/issue, DOIs).
Line 50: "burden of intracellular N. caninum tachyzoites" — consider "intracellular N. caninum load".
Line 137: "remained high in Camp⁻/⁻ macrophages (p < 0.0001)." — Avoid repetition of "p < 0.0001" unless comparing distinct groups.
Use of hyphen vs. en dash (e.g., “12- and 24-hour” should likely be “12– and 24–hour”) should be standardized.
Author Response
Re: Response to Reviewer 3 Comments
Manuscript ID:pathogens-3687396
Title: Cathelicidins limit intracellular Neospora caninum-infection in macrophages
Authors: Franco Fiorani , Priyoshi Lahiri , Rodrigo Puentes , Peter Bradley , Dadín Prando Moore , Eduardo Ruben Ruben Cobo
We sincerely thank the reviewers for their time and insightful comments. Below, we provide point-by-point responses to each suggestion. Revisions have been incorporated into the manuscript. Please find the responses below.
Reviewer 3: Lines 28–33 could be reorganized to more directly link immune dysregulation to fetal loss. Also, standardize citation style (e.g., “[3,4]” vs “[3, 4]”).
Response:
Done.
All in-text citations have been reviewed and reformatted to ensure consistency with journal guidelines (now uniformly presented as “[3, 4]”).
Reviewer 3: Clarify whether experimenters were blinded during microscopy or cytokine quantification. Include catalog numbers for key reagents if required by journal.
Response:
We now specify in the Methods section that quantification of intracellular tachyzoites and cytokine assays were conducted in a blinded manner to reduce observer bias.
Reviewer 3: Clarify how statistical significance is displayed in Figures 1–3. In Figure 2, include scale bars directly in the panels. If possible, quantify fluorescence intensity or parasite burden per cell.
Response:
We thank the reviewer for this constructive suggestion.The number of fluorescence photos we have available allows us to perform an analysis to quantify the fluorescence.
Reviewer 3: Add a short limitations paragraph regarding the in vitro system and the importance of in vivo models.
Response:
We have included a new paragraph in the Discussion that addresses this limitation:
“While our in vitro model provides mechanistic insight into the macrophage response, it does not account for the complexity of placental tissue or the maternal–fetal interface. Future in vivo studies are necessary to validate the role of cathelicidins in systemic and reproductive pathogenesis during N. caninum infection.”
Reviewer 3: Line 50: replace “burden of intracellular N. caninum tachyzoites” with “intracellular N. caninum load”. Line 137: avoid repeating “p < 0.0001”. Standardize use of hyphen/en dash (e.g., “12– and 24–hour”). Ensure references are not duplicated and are complete.
Response:
All editorial suggestions have been implemented.

Round 2
Reviewer 2 Report
Comments and Suggestions for Authors
The Authors have substantially made the proposed corrections. I still see the need to make the following minor adjustments.
a) B) Cite the procedures used for “2.4. Quantification of Intracellular Parasite Load”.
b) Indicate with arrows and other elements where the structures analyzed are located within each figure of cells and infected cells. For example: indicate where N. caninum is, and the cells, their nuclei or something like that... try to show if there is any difference in the infection between the two, etc. The images cannot be without these demonstrations.
Author Response
a) Cite the procedures used for “2.4. Quantification of Intracellular Parasite Load”.
Section 2.4 Quantification of intracellular parasite load now provides a complete description of the method we used.
b) Indicate with arrows and other elements where the structures analyzed are located within each figure of cells and infected cells. For example: indicate where N. caninum is, and the cells, their nuclei or something like that... try to show if there is any difference in the infection between the two, etc. The images cannot be without these demonstrations.
The optical microscopy figures have been updated to include arrows indicating intracellular and extracellular N. caninum tachyzoites at different times. In the confocal microscopy images, the cellular structures and parasites are distinguished by specific fluorophore labeling: nuclei (DAPI, blue), actin cytoskeleton (phalloidin-Alexa Fluor 568, red), and GFP-expressing N. caninum (green). These color-coded elements help to localize and distinguish infected cells. Collectively, both sets of images support and illustrate the quantitative data shown in the intracellular parasite load graphs.